# A Window into New Insights on Progression Independent of Relapse Activity in Multiple Sclerosis: Role of Therapies and Current Perspective

**DOI:** 10.3390/ijms26030884

**Published:** 2025-01-21

**Authors:** Tommaso Guerra, Pietro Iaffaldano

**Affiliations:** Department of Translational Biomedicine and Neurosciences (DiBraiN), University of Bari “Aldo Moro”, 70121 Bari, Italy; guerra.tommaso93@gmail.com

**Keywords:** multiple sclerosis, PIRA, smoldering disease

## Abstract

In multiple sclerosis (MS), there is significant evidence indicating that both progression independent of relapse activity (PIRA) and relapse-related worsening events contribute to the accumulation of progressive disability from the onset of the disease and throughout its course. Understanding the compartmentalized pathophysiology of MS would enhance comprehension of disease progression mechanisms, overcoming the traditional distinction in phenotypes. Smoldering MS activity is thought to be maintained by a continuous interaction between the parenchymal chronic processes of neuroinflammation and neurodegeneration and the intrathecal compartment. This review provides a comprehensive and up-to-date overview of the neuropathological and immunological evidence related to the mechanisms underlying PIRA phenomena in MS, with a focus on studies investigating the impact of currently available therapies on these complex mechanisms.

## 1. Introduction

For decades, neurologists have struggled to assess and anticipate the unpredictable and variable clinical course of multiple sclerosis (MS), much like a challenging riddle that defies an easy resolution. What is the basis of smoldering MS activity? How can the continuum of neurodegeneration and neuroinflammation be described and treated? Recently, the literature has been moving towards solving these questions.

MS is traditionally classified as relapsing–remitting (RR) or primary progressive (PP), based on the initial disease course; over the time of disability accrual, a secondary progressive (SP) MS phenotype can be developed [1,2]. MS patients often continue to report the worsening of symptoms even while the peripheral immune response is being well controlled with disease-modifying treatments (DMTs) [3,4]; deciphering compartmentalized pathophysiology in MS would, therefore, aid a deeper comprehension of mechanisms of disease progression, overcoming the classic distinction in phenotypes [5,6,7,8]. Both progression independent of relapse activity (PIRA) and relapse-related worsening (RAW) events contribute to the accumulation of irreversible neurological impairment since the disease onset and throughout its course [9,10,11].

To harmonize the definition of PIRA, it has been suggested that an increased Expanded Disability Status Scale (EDSS) score at least three months after and one month prior to the onset of a reported relapse, a new baseline score following each relapse, and a confirmation score at least three months after the reported disability worsening and one month before the onset of a described relapse are considered [12,13]. Debate has surrounded this clinically based definition, which has both strengths and limitations [13], and which is heterogeneously described in real-world cohorts [14]. Recently, the definition of PIRMA, progression independent of relapse and magnetic resonance imaging (MRI) activity, was also coined to expand the clinical definition of PIRA including the absence of MRI activity between three months after and one month before the onset of a relapse [13].

The research into the etiological mechanisms of PIRA is currently spurred on by the finding that it might manifest early in MS patients [15]. PIRA constitutes the main driver of disability accrual, and it is associated with higher disability in the long term [15,16,17]. RAW represents a less frequent phenomenon, likely due to the effectiveness of treatments on neuroinflammation-related phenomena [18]. MS patients presenting with PIRA after a first demyelinating event have an unfavorable long-term prognosis, with a higher probability of and a faster progression rate to disability milestones, especially if PIRA occurs early in the disease’s course [19]. A longer disease duration and an older age at onset were factors linked to PIRA: between the ages of 21 and 30, the frequency of PIRA almost doubled, and between the ages of 40 and 70, it increased by around seven times in a recent study [15]. Compared to patients without PIRA, individuals with PIRA experience a noticeably higher rise in EDSS scores, and PIRA’s impact on hastening the progression of EDSS is less noticeable in pediatric-onset MS than in adult-onset and late-onset MS [17]. Patients with a relapsing course and PIRA also experience rapid brain atrophy, especially in the cerebral cortex [20].

Heterogeneous biomarkers of progression in MS have been proposed over the last few years [21]. In MS, inflammation may be the initial cause of neurodegeneration, which is then sustained by the ongoing compartmentalized inflammation [6]. Smoldering MS activity is supposed to be sustained by an ongoing dynamic between the intrathecal compartment, comprising cerebrospinal fluid (CSF) and meninges, and the parenchymal chronic processes of inflammation and neurodegeneration. This persistent interaction promotes glial activation and neuron loss, caused by a crucial imbalance between damage, repair, and brain functional reserve [22,23,24].

With emphasis on the evidence regarding the impact of currently available therapies on these multifaceted mechanisms, this review attempts to give a broad overview of current understandings of neuropathological and immunological evidence regarding the mechanisms causing PIRA phenomena in MS.

## 2. Methodology

### 2.1. Aims and Research Planning

This review aims to provide a comprehensive and up-to-date overview of the neuropathological evidence related to the mechanisms underlying PIRA phenomena in MS, with particular reference to studies investigating the impact of currently available DMTs on these phenomena. We carefully scrutinized the MS literature, examining and discussing the numerous neuroimmunology topics in the different review thematic sections.

### 2.2. Search Strategy and Inclusion and Exclusion Criteria

A specific approach for selecting literature sources has been applied in the two primary academic databases, PubMed and Google Scholar. Our search strategy comprised pertinent keywords related to PIRA and smoldering disease in MS. During the selection procedure, particular inclusion and exclusion criteria were applied to guarantee the high standard and applicability of this review. We considered the most recent PIRA-focused research articles or reviews; studies offering insights into pathophysiology and immunology implications of MS and related to disease progression mechanisms; and clinical trials and studies of disease registers and real-world cohorts on therapeutic targets in MS in terms of neuroinflammation and neurodegeneration. In two cases, we also included topics discussed at the most important European congress on MS research, the European Committee for Treatment and Research in Multiple Sclerosis. Following this search path, some papers were eliminated since they were either unpublished manuscripts or non-peer-reviewed materials. Specifying the time frame of the search, no restrictions were placed on publication dates, but we focused mainly on studies published in the last five years. Thus, about 85% of citations retrieved are from studies published after 2020, and about 32.5% are from research published in 2024.

## 3. Smoldering Biology in Multiple Sclerosis

Even with steady inflammatory parameters, people with MS frequently see a worsening of their motor and cognitive impairment. Smoldering-associated worsening is thought to be caused by a confluence of degenerative and inflammatory mechanisms, including anterograde and retrograde axonal degeneration, combined with the breakdown of the compensatory processes of neuronal plasticity and remyelination [25]. PIRA can be considered an epiphenomenon of smoldering MS disease, but research is limited to well-validated MS outcome measures in clinical trials and real-world studies [26].

The field of MS biomarkers has changed considerably in recent years due to significant improvements in assay technologies and a focused attempt to identify molecules that may provide information on these pathogenic mechanisms [6,27]. Recent positron emission tomography (PET) studies employing radioligands for innate immunity assessment demonstrated that a smoldering component, which predicts cortical atrophy and EDSS progression, is present in a surprisingly high percentage of MS lesions [28]. Absinta et al. defined the primary roles, in the neurodegenerative programming, of “microglia inflamed in MS” (MIMS) and “astrocytes inflamed in MS”. Genes linked to foam-cell differentiation, lipid storage, lipoprotein-related pathways, lysosome metabolism, and inflammatory response regulation all resulted upregulated in the "MIMS-foamy" pattern. In parallel, “MIMS-iron” was characterized by the overexpression of ribosomal protein-encoding genes, resulting in the activation and expression of the MHC class II protein complex. MIMS-iron analysis also revealed an upregulation of ferritin, immunoglobulin Fcγ, and complement component C1 complex alongside the highest expression of the inflammatory cytokine gene IL1B. The expression of MHC-related and inflammatory markers—such as SOD1, iron-related genes FTH1 and FTL, and C1-complex genes—was higher in MIMS-iron when compared directly to MIMS-foamy, suggesting a fundamental role in antigen presentation and the spreading of inflammatory damage at the lesion edge [29]. An unbiased in silico approach was able to identify a convergence of cross-regional transcriptomic reactivity in MS oligodendrocytes, leading to an upregulation of the myelin-encoding gene MAG and downregulation of the muscarinic receptor-encoding gene CHRM5, contributing to an unresolved self-sustaining inflammation [30].

According to neuro-molecular research on progressive MS, chemokines and cytokines generated by circulating immune cells and meningeal tertiary lymphoid structures may permeate the CSF and enter the cortex, activating microglia and causing damage [31,32].

Mitochondrial dysfunction has also been pointed out as a putative mechanism underlying PIRA. The development of the neurodegeneration molecular cascade is significantly influenced by nuclear factor erythroid 2-related factor 2 (Nrf2)-impaired signaling, which is fundamental to preventing oxidative stress and mitochondrial dysfunction and which is found to be compromised in patients with MS [33,34]. It is possible for mitochondrial disfunction to be triggered by reactive oxygen species (ROS) produced by innate immune cells, spurring the process of axonal swelling and fragmentation in disease progression [6,35]. Furthermore, both reactive astrocytes and chronically demyelinated axons have a higher quantity of mitochondria because demyelination raises the energy requirement to maintain a proper intra-axonal ion balance [36].

Abnormalities in the glymphatic system, a complex CNS “waste clearance” system, have been reported in several neurodegenerative diseases, including MS. Impaired glymphatic function in MS was associated with measures of neurodegeneration and demyelination, in a proposed mutual relation: the glymphatic system dysregulation exacerbates MS symptoms and, in parallel, MS neuroinflammation impacts the correct functioning of the system [37,38]. Mechanisms of inflammation and demyelization can disrupt astrocyte function, which is essential to glymphatic system activity [39]. One of the connections between MS and glymphatic dysfunctions may also be the presence of inflammatory lesions in the perivenous spaces, a drainage route through which the glymphatic system eliminates waste products [38]. Carotenuto et al. [37] demonstrated that the glymphatic function was usually compromised in MS patients when compared to healthy controls, and PPMS patients showed the most severe impairment.

## 4. Radiological Expression of PIRA and Molecular Correlates

Chronic active lesions (CAL) are a crucial sign of chronic inflammation. A recent consensus established biomarkers of CAL, including paramagnetic rim lesions (PRL) identified on susceptibility-sensitive MRI, MRI-defined slowly expanding lesions (SELs), and 18-kDa translocator protein (TSPO)-positive lesions on PET. In this complex assessment of biomarkers, the one with the strongest histological and molecular support is PRL [40]. Histologically, the cellular substrate of iron-positive rim lesions consists of iron-loaded activated myeloid cells and related pathways [41]. Hofman et al. observed the upregulation of the CD163-HMOX1-HAMP axis at the rims of chronic active lesions, indicating that haptoglobin-bound hemoglobin is the crucial source of MC-associated iron uptake, which is also confirmed by the strong association between PRL levels in MS and CSF-associated sCD163 [42]. Additionally, the C1QA, HMOX1, and HAMP genes were activated in those cells, constituting a molecular sign of a pro-inflammatory profile. On the other hand, IL10 mRNA was elevated in perilesional myeloid cells, which may indicate tissue-regulating and anti-inflammatory properties in different areas of the rim [42].

PRLs have been linked to a more severe course of the disease [43], and to elevated levels of serum neurofilament light chain (sNfL) [44], without a clear correlation with their topographical distribution [45]. The number of PRL was also associated with the number of leptomeningeal contrast enhancement foci on T2-FLAIR and real-reconstruction inversion recovery, linking leptomeninges to mechanisms related to the sustaining chronic inflammation [46].

SELs represent the subset of non-enhancing chronic lesions showing radial and linear expansion over 1–2 years. The proportion of SELs has been correlated with MS progression after 9 years, considering, in addition, severe SEL microstructural abnormalities as a predictor of EDSS worsening and SPMS conversion [47].

Both diffuse and localized neurodegenerative processes throughout the CNS appear to be important factors linked to the development of PIRA, according to a recent large multicentric study by Cagol et al. [48]. PRLs have also been linked to higher rates of brain and spinal cord atrophy [49,50].

Considering the key role of meningeal inflammation in MS pathogenesis and progression mechanisms, the findings of Herranz et al. strongly support the role of TSPO-PET for imaging in vivo cortico-meningeal inflammation [51].

A reduced diffusion tensor imaging along the perivascular space (DTI-ALPS) index has recently been correlated to SPMS conversion, focusing on the possible key role of microstructural glymphatic disruptions in MS progression mechanisms [52].

## 5. Meningeal Inflammation, Subpial Cortical Damage, and Focus on Microglia and Diffuse White Matter Pathology

One of the major contributors to the pathophysiology of cortical demyelination in MS is believed to be compartmentalized meningeal inflammation [53]. A change in the balance of tumor necrosis factor (TNF) signaling from TNFR1/TNFR2 and NFkB-mediated anti-apoptotic pathways to pro-apoptotic/pro-necroptotic signaling mediated by TNFR1 and RIPK3 in the grey matter (GM) was linked to increased meningeal inflammation. In the GM of the MS cortex, TNFR1 was shown to be mainly expressed in neurons and oligodendrocytes, while TNFR2 was primarily expressed in microglia and astrocytes [54].

A recent study characterized the cortical and meningeal translocator protein (TSPO) expression in vivo and ex vivo. The TSPO signal was abnormally elevated in the meningeal tissue and cortex of MS patients, diffusively in progressive and localized in relapsing phenotypes. In post-mortem SPMS, immunohistochemistry disclosed increased TSPO expression in the meninges and adjacent subpial cortical lesions, which is related to meningeal inflammation. Meningeal MHC-class II+ macrophages and cortical-activated MHC-class II+ TMEM119+ microglia were found to exhibit translocator protein immunostaining [51].

Activated microglia and infiltrating macrophages may become iron-loaded in actively demyelinating MS lesions [55]. Post-mortem results suggest that early lesion formation may be attributed to MS microglia nodules. In the MS microglia nodules’ environment, an upregulation of genes related to phagocytosis, adaptive and innate immune responses, lymphocyte activation, lipid metabolism, and metabolic stress has been observed. Van den Bosch et al. also found significant activation of complement, IgG transcription, and MAC formation, constituting a hypermetabolic state with elevated pro-inflammatory cytokines and expression of ROS genes. The close proximity of activated lymphocytes that are impacting and being influenced by the microglia nodules is revealed by gene expression analysis: NCKAP1L, CASP3, JAK3, TCIRG1, CORO1A, GRB2, IRF8, TLR2, and IL18 [56].

Compared to healthy controls, MS patients have greater levels of innate inflammation in their normal-appearing white matter (NAWM) and cortex [28,57]. Therefore, the accumulation of focal lesions and microstructural tissue abnormalities in cognitively linked WM tracts, the occurrence of focal and diffuse damage in important GM regions, and the presence of functional brain network abnormalities are the causes of the so-called “disconnection syndrome”, whose clinical correlation is impaired cognition and fatigue [58].

A recent review by Zhan and colleagues [59] outlined the role of ectopic lymphoid follicles in MS, stressing the correlation with cortical [60] and even spinal cord [61] pathology. The inflammatory meningeal and perivascular infiltrates were shown to contain a large number of CXCR5+ cells, cytoplasmic nuclear factor of activated T-cell-positive (NFATc1+) cells, enriched CD3+CD27+ memory cells, and CD4+CD69+ tissue-resident cells [62].

The main biological processes described, with a focus on CAL, paramagnetic rim, and meningeal and cortical inflammation mechanisms, are shown in Figure 1.

## 6. Adaptive Immunity and PIRA: Role of T Cells and B Cells

In MS pathogenesis, B cells are crucial not only for antibody-dependent mechanisms but also, through the abnormal production of cytokine and chemokine and an antigen-presenting function that activates T cells and drives autoproliferation of brain-homing T cells, for exacerbating the non-resolving neuroinflammation [63,64]. Furthermore, the contribution of B cells to the formation of ectopic lymphoid aggregates in the meninges is largely described [59]. B cell-depleting therapies have significantly contributed to modifying the natural history of MS [65,66].

The intricate interplay between extrinsic mechanisms, requiring regulatory T cells (Tregs), and T-cell specific costimulators regulates peripheral tolerance. The importance of regulatory molecules and their associated receptors in peripheral T-cell tolerance and T-cell function, and their activation pathways in MS have been extensively outlined in a recent review [67]. MS has been linked to disruptions in the expression of various costimulatory signaling molecules necessary for T-cell activation [68]. The interaction of programmed cell death 1 (PD-1) and its ligands results in the start of inhibitory signals that control tissue damage and T-cell activation [69]. It has been confirmed that the PD-1/PD-L1 axis is a classical immune suppressor in MS, and despite the regulation mechanisms of this axis throughout the MS course still being unknown, several studies have shown PD-L1 deficiency and a correlation with disease progression [70]. CD137 has been proposed to be involved in the defective regulatory function of Treg and dendritic cells and resulted elevated in MS individuals, while FoxP3 was found to be impaired in relapsing phenotypes [68]. B7H4, VISTA, CTLA-4, BTLA, Lag-3, and TIM3 receptors and their ligands also have a key role in peripheral tolerance and an inhibitory effect on cytokine production [67].

What is there to say regarding the relation between the adaptative immune response and PIRA? Planas et al. [71] used a high-throughput T cell receptor β-chain variable gene (TRBV) sequencing of genomic DNA to characterize WM demyelinating lesions in SPMS patients. They highlighted a significant sharing of clonotypes of T cells, clonally expanded T cells with the same TRBV sequence, across MS lesions regardless of their proximity and independently of inflammatory activity, alongside specific brain homing of CD4+. T and B cell meningeal accumulations were located next to subpial cortical lesions, and larger subpial lesion regions were linked to higher immune cell accumulation. In patients with significant meningeal inflammation, the percentage of active and mixed active/inactive WM lesions was higher, while the percentage of inactive and remyelinated WM lesions was generally lower [72].

Fransen et al. found an increased number of T cells clustering in the perivascular space and, in MS lesions with active and inactive features, identified a specific tissue-resident memory phenotype of CD8+ T cells that lack S1P1 and express CD69, CD103, CD44, CD49a, PD-1, in parallel with upregulated markers for homing (CXCR6), reactivation (Ki-67), and cytotoxicity (GPR56) [73].

## 7. Fluid Biomarkers and PIRA Phenomena

CSF biomarkers linked to immune-related pathways predict future impairment and correlate with clinical and imaging outcomes of MS severity [6,74].

A variety of immunoassays can be used to quantify NfL in both CSF and blood, which is released in the CNS’s interstitial space when axonal damage occurs [75]. In addition to a clear correlation with disease activity parameters [76], Abdelhak et al. recently documented the occurrence of NfL elevation in advance of confirmed disability worsening independent of clinical relapses [77].

A new triggering role of parvalbumin, a calcium homeostasis-regulating protein expressed by particular subsets of fast-spiking GABAergic interneurons, has emerged in recent years. In postmortem MS, parvalbumin levels in the CSF indicate the loss of cortical neurons and correlate at baseline with the cortical atrophy of specific brain regions that are known to be particularly impacted by cortical disease [78]. A possible prognostic neurodegenerative biomarker of GM dysfunction was suggested to be parvalbumin levels in the CSF at the time of MS diagnosis. A correlation has been found with the volume of the following regions: inferior frontal and postcentral gyrus, frontal pole, transverse temporal gyrus, cerebellar cortex, right thalamus, pericalcarine cortex, lingual gyrus, and medial frontal gyrus. In parallel, a clinical correlation with cognitive impairment, physical disability, and fatigue has been found [79].

Cross et al.’s research aimed to identify markers of non-relapsing progressive biology, disentangling them from markers of acute inflammatory relapse biology. Levels of glial fibrillary acidic protein (GFAP), a protein highly expressed by astrocytes, were not linked to acute inflammatory measures but correlated with SEL count, a greater proportion of T2 lesion volume from SELs, and lower T1-weighted intensity within SELs; neurofilament heavy chain was correlated with SEL count and lower T1-weighted intensity within SELs [80]. Differently from serum levels of NfL, serum GFAP concentration does not typically elevate during acute inflammation but reflects accelerated GM brain volume loss and can be considered a prognostic biomarker for future PIRA [81].

The critical significance of CSF cytokines and their connection to cortical disease have recently come to light [31,82]. High levels of CSF chemokines are strongly related to lymphoid neogenesis and B cells with cortical damage accumulation over the long term [83].

Compared to non-inflammatory control patients, subjects with progressive MS had higher CSF concentrations of free-circulating mitochondrial DNA (mtDNA) copies, which positively correlated with EDSS worsening and a higher T2 lesion load [84]. Therefore, elevated levels of cell-free mtDNA were linked to disease progression mechanisms, stressing the key role of mitochondria. CSF levels of lactate are also related to mitochondrial dysfunction and were linked to an increase in neurological impairment and to levels of both tau and neurofilament light protein levels [85].

## 8. Therapies and PIRA

Determining how DMTs affect PIRA can help unravel the complexity of silent progression pathways. A decrease in relapse activity and, thus, RAW, as well as a reduction in the formation of new MRI lesions, which may eventually reduce the risk of PIRA, were the main explanations for therapeutic benefit in trials investigating PIRA [9,13,86]. PIRA events can be observed during treatment with high-efficacy (HE) DMTs, which completely reduce clinical and radiological disease activity [87]. Therefore, HE DMTs decrease relapse-related disability, increasing the probability that other observed events of impairment accrual are unrelated to relapses. This embodies a phenomenon of “unmasking”: therapies currently in use are excellent at reducing events strictly related to neuroinflammation, which leads to the defining of the remaining progression events as dependent on other pathways, especially neurodegenerative silent progression.

The time interval between disease onset and the start of the first DMT is a well-recognized strong predictor of disability accumulation, independent of relapse activity, over the long term [88]. A delayed DMT initiation [15] and a shorter treatment exposure [89] are associated with a higher risk of both PIRA and RAW events.

Compartmentalized inflammation in the CNS, reflected by the presence of PRLs, seems to be stubbornly resistant to the DMTs currently in use [43,90]. The different pathways of action of DMTs used in clinical practice, however, may have a molecular and biological, and hence clinical, impact on the mechanisms associated with “silent progression”, as some data in the literature have demonstrated. Considering the clinical perspective, results of a study from the database of the Italian Multiple Sclerosis Register [91] demonstrated that both ocrelizumab and natalizumab strongly suppress RAW events and have a similar impact on PIRA in naïve RRMS patients [92]. PIRA was never observed in pediatric-onset MS and was reported in a small percentage of adult-onset MS patients during the follow-up (40.0 ± 25.9 months) of a natalizumab-treated cohort in a recent study [93]. Comparing the efficacy of natalizumab with platform therapies in SPMS, the proportion of patients who developed PIRA at 48 months was significantly higher in the interferon beta-1b group compared to the natalizumab-treated cohort (72.4% versus 40.2%, *p* = 0.01) [94]. The molecular effectiveness of anti-CD20 monoclonal antibodies [95] is also reflected in the CSF measures of lymphocyte biology (sTACI, sCD27, and sBCMA) and chemokines (CXCL10 and CXCL12) [80].

The effect of DMTs on neurodegenerative mechanisms can be unraveled by observing variations in brain volume loss and CAL, epiphenomena of persistent compartmentalized inflammation, and neurodegeneration. Compared to fingolimod, ocrelizumab-treated patients in this study experienced fewer new white matter lesions, lower deep GM volume loss, and reduced cortical thinning globally and in several specific regions of interest [96]. Using a new post-processing MRI method called T1/T2 ratio of iron rim lesions, Eisele et al. found that patients on fingolimod, dimethyl fumarate, and ocrelizumab had a significantly lower 2-year follow-up rate than those not on DMTs. Their findings suggest that DMTs might have a slightly beneficial long-term effect on smoldering MS lesions [97]. Measures of chronic lesion activity were lowered by ocrelizumab: mean normalized T1 signal intensity and T1 hypointense lesion volume accumulation decreased in both slowly and non-slowly expanding lesions [98]. Despite the predicted effects on inflammatory networks related to microglia in CAL, anti-CD20 monoclonal antibodies failed to fully resolve paramagnetic rim lesions after a 2-year MRI follow-up, probably due to the paucity of B cells in CAL, ineffective transit of anti-CD20 antibodies across the blood–brain barrier, and limited tissue turnover of B cells [99]. Furthermore, exploring the effect of DMTs on SELs, Preziosa et al. recently found that the effects of natalizumab and fingolimod on SEL occurrence were modest, with natalizumab being slightly more effective [100].

Analyzing the comparison of patients started on low-to-moderate efficacy therapies (LM-DMT) and those who received first-line HE-DMT in the Swedish MS Register, Spelman et al. found a significantly higher unadjusted rate of CDW events in the LM-efficacy group. The HE-DMT group also had a lower rate of PIRA, although the difference between the two groups in this measure was not statistically significant [101]. The relative contributions of PIRA and RAW to the evolution of EDSS in individuals diagnosed and treated at different times were examined in a recent Italian research on 1405 patients followed for an average of 14.3 years, emphasizing a deceleration of the MS’s course throughout the years, as determined not only by fewer RAW events, but also by a reduction in PIRA, as a result of DMT use. In patients diagnosed in 1980–1996 and 1997–2008, PIRA’s average contribution to the overall advancement of EDSS was already significant; however, in patients diagnosed in subsequent years, this contribution substantially increased [102]. Also considering the most recently approved DMTs, ofatumumab markedly reduced the risk of PIRA in early RMS patients in the pooled ASCLEPIOS population, compared to teriflunomide [103]. Subjects treated with cladribine exhibited an effect on neurodegenerative MRI biomarkers, with markedly reduced annualized brain atrophy [104] and GM volume loss [105] rates compared to those treated with a placebo [106]. The references mentioned in this paragraph are better characterized in Table 1.

BTK inhibitors (BTKi) represent a potentially effective therapeutic strategy to prevent PIRA and smoldering MS [107]. BTK activity is essential for B cell maturation and function as well as for B cell and myeloid cell intracellular signaling, including microglial pathways. Consequently, BTK inhibition causes peripheral B cell regulation, maturation, proliferation, and autoantibody and cytokine production, along with a decrease in macrophage and microglial activity in the central nervous system [108,109]. Tolebrutinib, which demonstrated greater CNS penetration compared to other BTKi, reduced the volume of SELs [110], modulated microglial genes, and had non-cell autonomous impacts on neurons and astrocytes during stimulation of fragment crystallizable gamma receptors [111].

It is evident from the molecular pathways previously revealed and currently being investigated that new treatment strategies for MS are needed, along with unconventional thinking [112].

A “neuro-lympho-vascular component” in neurodegenerative conditions has been proposed in the last year [113]. A recent study of das Neves et al. highlighted that reduced vascular endothelial growth factor C in the CSF can be observed in MS patients, especially soon after clinical relapses, constituting a possible sign of impaired meningeal lymphatic function. In light of this, and of the fact that meningeal lymphatics control oligodendrocyte function and brain myelination, this study found that inducing ablation of the meningeal lymphatic vessels in adult mice resulted in changes in glial cell gene expression [114].

Novel therapies that improve the function of oligodendrocytes and other glial cells may be useful to prevent neurodegeneration and repair structural damage, in addition to altering or enhancing several metabolic pathways.

**Table 1 ijms-26-00884-t001:** Investigations into how disease-modifying therapies affect PIRA.

Authors	Study Design	Population	Interpretation of Results
Graf et al. [87]	Retrospective chart review study	184 RRMS patients	Patients who are started on natalizumab early in the course of their disease, typically to treat an aggressive clinical presentation, are more likely to experience early confirmed progression independent of relapse activity.
Iaffaldano et al. [88]	Retrospective cohort study	11,871 MS patients (BMSD)	DMTs should be commenced within 1.2 years from the disease onset to reduce the risk of disability accumulation over the long term.
Portaccio et al. [89]	Retrospective cohort study	5169 MS patients (CIS, RRMS) (RISM)	Longer exposure to DMT is associated with a lower risk of both progression independent of relapse activity and relapse-associated worsening events.
Iaffaldano et al. [92]	Retrospective cohort study	Total population: 770 MS patients. Matched cohort: 195 patients treated with ocrelizumab, 195 with natalizumab (RISM)	Natalizumab and ocrelizumab strongly suppress RAW events and, in the short term, the risk of achieving PIRA events, EDSS 4.0 and 6.0 disability milestones is not significantly different.
Puthenparampil et al. [93]	Observational retrospective study	Total population: 160 MS patients. Matched cohort: 32 patients pediatric-onset MS and 64 with adult-onset MS	In naïve patients treated with natalizumab, PIRA was never observed in pediatric-onset MS, while a small percentage of adult-onset MS (12.5%) had PIRA events during the follow-up.
Chisari et al. [94]	Retrospective cohort study	Total population: 5321 SPMS patients. Matched cohort: 421 MS patients treated with natalizumab and 353 with interferon-beta 1b (RISM)	The proportion of patients who developed PIRA at 48 months is significantly higher in interferon beta-1b group compared to the natalizumab-treated cohort. Patients treated with IFNb-1b are 1.64 times more to likely to develop PIRA
Cross et al. [80]	Cohort study assessed data from 2 prospective MS cohorts	Test cohort: 131 MS patients Confirmation cohort: 68 MS patients.	Ocrelizumab reduced CSF measures of acute inflammation, including lymphocyte measures sTACI, sCD27, sBCMA, and chemokine/cytokine measures CXCL13 and CXCL10. Neuroaxonal injury measure NfH and glial measures sTREM2 and YKL-40 resulted modestly reduced.
Bajrami et al. [96]	Observational, prospective, longitudinal study	95 RRMS	Compared to fingolimod, ocrelizumab-treated patients experience fewer new white matter lesions and lower deep grey matter volume loss, lower global cortical thickness change, and reduced cortical thinning/volume loss in several regions of interest.
Eisele et al. [97]	Retrospective study	27 MS patients	Patients on fingolimod, dimethyl fumarate, and ocrelizumab have a considerably lower 2-year follow-up rate of T1/T2 ratio of iron rim lesions. than those not taking DMTs.
Elliott et al. [98]	PPMS study population of the ORATORIO trial	ITT population (*n* = 732); SEL analytical population (*n* = 555)	Ocrelizumab reduces longitudinal measures of chronic lesion activity such as T1 hypointense lesion volume accumulation and mean normalized T1 signal intensity decrease both in slowly expanding/evolving and non-slowly expanding/evolving lesions.
Maggi et al. [99]	Retrospective analysis and imaging, laboratory, and clinical data prospectively collected	72 MS patients	Despite predicted effects on inflammatory networks related to microglia in CAL, anti-CD20 monoclonal antibodies failed to fully resolve paramagnetic rim lesions after a 2-year MRI follow-up
Preziosa et al. [100]	Single centre, prospective, longitudinal, open-label, non-randomized cohort study	52 MS patients	Higher SEL number and volume is observed in the fingolimod vs. natalizumab group. Longitudinally, non-SEL MTR increased in both treatment groups. *T*_1_ signal intensity decreased in SELs with both treatments and increased in natalizumab non-SELs.
Montobbio et al. [102]	Retrospective study	1405 MS patients	Across ages, patients diagnosed in more recent times had lower PIRA and RAW than those diagnosed in earlier periods. Patients diagnosed in later years had a significantly higher contribution of PIRA in EDSS progression.
Cortese et al. [105]	MRI data from the CLARITY study	Treatment group: 267 MS patients Placebo group: 265 MS patients	In the first six months of treatment, patients on cladribine experienced more GM and WM volume loss than those on placebo, most likely as a result of pseudoatrophy. Nonetheless, GM volume loss was considerably less in cladribine-treated patients than in placebo-treated group throughout the course of 6–24 months.

Abbreviations: BMSD, Big Multiple Sclerosis Data Network; DMT, disease-modifying therapies; GM, gray matter; MS, multiple sclerosis; MTR, magnetization transfer ratio; MRI, magnetic resonance imaging; PIRA, progression independent of relapse activity; RISM, Italian Multiple Sclerosis Register; ITT, intention to treat; RAW, relapse associated worsening; SELs, slowly expanding lesions; and WM, white matter.

## 9. Conclusions and Future Perspectives

Neurologists and researchers are constantly trying to solve the pathologic puzzle of PIRA, which is essential to understanding and tackling neurodegeneration in MS [115]. Some research grounds are still to be explored. Since the glymphatic system’s role in MS progression is still a relatively young field of study, additional research is required to fully comprehend the nature of this mutual relationship and any potential therapeutic applications [37,38]. A solution to halting non-relapse-related progression in MS patients could lie in understanding how aging affects immune and brain cell activity [116]. Further studies are needed in these directions [117,118].

Understanding the phenomena of neurodegeneration and better characterizing the mechanisms underlying the definition of PIRA are two of the many challenges in the neuroimaging research field. The term “advanced-PIRMA” has been recently proposed by Ciccarelli et al. to stress the advocacy of employing both conventional and advanced imaging assessment to examine the contributions to disability accumulation of all the underlying pathologic processes, considering new lesions, PRLs, SELs, atrophy, and all neurodegenerative radiological parameters [13]. Artificial intelligence and machine learning approaches have the potential to improve MS research: by combining data from many sources, these methods might predict clinical impairment and future disease progression, while also generating measures that were previously only achievable with specialized imaging (such as synthesizing new images from conventional sequences) [119].

In light of the first insights into the new diagnostic criteria for MS proposed at ECTRIMS 2024 [120], disease biomarkers play the main role in defining and estimating biological damage, especially for radiologically isolated syndrome. A radical change in the approach to the disease is also necessary in clinical terms. Neurologists should concentrate on identifying subtle neurological impairments in cognitive and physical functioning, and in this way expand the scope of EDSS. Management focused on increasing knowledge of the disease’s progression, which starts at the time of MS diagnosis, and on how progressive neurodegeneration phenomena are linked to a disease course that is only apparently stable should also be a priority. Outcomes in clinical trials should be improved by including PIRA-related biological correlations, for example, the evolution/resolution of PRLs. Assessing meningeal inflammation, as indicated by various analytes, whenever feasible, may suggest the possibility of therapeutic benefit [90]. Disease registries should be expanded to include variables derived from the study of more recent metabolic patterns or should be easier to link to genetic and histopathological databases [121]. Therefore, the call for studies aimed at collecting clinical, genetic, and imaging data together is crucial, encouraging the idea of integrating datasets to define a multidimensional picture of MS patients [122].

Patients with onset at different ages have very deeply heterogeneous clinical profiles and outcomes, which highlights the need for “age-specific” MS management approaches, especially for late-onset patients [123].

In conclusion, novel perspectives on the biology and progression of MS have helped to promote and confirm the idea of a smoldering MS caused by an inextricable tangle of acute peripheral inflammation, chronic neuroinflammation, and neurodegeneration mechanisms. A “biological profiling” of the patient will be necessary, in parallel with the ongoing abolition of classic MS disease phenotypes.

## Figures and Tables

**Figure 1 ijms-26-00884-f001:**
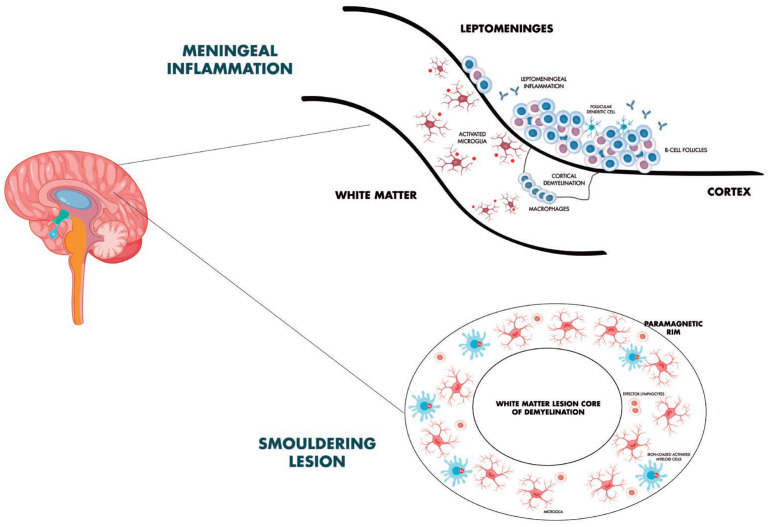
Smoldering biology in multiple sclerosis: focus on chronic active lesions, meningeal inflammation, and subpial cortical damage.

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
