# Peer review of "A Window into New Insights on Progression Independent of Relapse Activity in Multiple Sclerosis: Role of Therapies and Current Perspective"

_ijms, 2025, doi:10.3390/ijms26030884_

Round 1
Reviewer 1 Report
Comments and Suggestions for Authors
In the review article „A Window to New Insights on Progression Independent of Relapse Activity in Multiple Sclerosis: Role of Therapies and Current Perspective“, the authors provide an overview of the mechanisms underlying progression independent of relapse activity (PIRA) in multiple sclerosis and focus on the impact of disease-modifying treatments on PIRA. Although the authors have addressed the current topic in an informative and systematic manner, there are some issues that require attention:
What are the authors’ views on the hypothesis that disease-modifying treatment with high efficacy on relapses may increase the number of PIRA events compared to placebo? I recommend discussing this issue in section 8.
Mitochondrial dysfunction as a putative mechanism underlying PIRA should be mentioned.
Given that aging has a major impact on the pathological and immunological processes in MS and that there has been a recent shift in MS prevalence towards older age and also a marked increase in the proportion of patients with late-onset MS, it would be useful to present data on PIRA in late-onset MS, if such data exist.
It would also be useful to explain the term PIRMA
Author Response
Response to Reviewer 1 Comments
The reviewer's time spent reviewing this manuscript is greatly appreciated. Please find the detailed responses below and the corresponding revisions highlighted in track changes in the re-submitted files.
Comment 1: What are the authors’ views on the hypothesis that disease-modifying treatment with high efficacy on relapses may increase the number of PIRA events compared to placebo? I recommend discussing this issue in section 8.
We sincerely thank the reviewer for the scrutiny of our review, allowing us to better discuss this challenging topic. We can talk, in our opinion and in line with the latest expert reviews, of a phenomenon of “unmasking”: the therapies currently in use are excellent at reducing events strictly related to a high neuroinflammation, which therefore leads to define the remaining progression events as dependent on other phenomena, especially neurodegenerative. Ciccarelli et al. (Neurology 2024) accurately addressed this topic in their paper, discussing limitations of PIRA definition in clinical trials. Relapse-related disability is decreased by a treatment that suppresses relapse activity, increasing the probability that the other observed events of impairment accrual are unrelated to relapses. Given that EDSS increases are more likely to be linked to relapses that are more frequent in the placebo arm, a recent simulation of a randomized controlled trial that was proposed at ECTRIMS 2023 and supported this theory by demonstrating that the placebo arm experiences fewer PIRA events than the treatment arm. Therefore, if PIRA is to be employed as a major end point, accurate detection of all RAW occurrences is required. A treatment that is highly effective at preventing relapses may also artificially increase the frequency of PIRA events as compared to placebo. We have included this topic in our section 8, with a better definition of the unmasking phenomenon. Determining how DMTs affect PIRA can help unravel the complexity of silent progression pathways. A decrease in relapse activity and, thus, RAW, as well as a reduction of the formation of new MRI lesions, which may eventually reduce the risk of PIRA, were the main explanations for therapeutic benefit in trials investigating PIRA [9, 13, 86]. PIRA events can be observed during treatment with high-efficacy (HE) DMTs, which completely reduce clinical and radiological disease activity [87]. Therefore, HE DMTs decrease re-lapse-related disability, increasing the probability that other observed events of impairment accrual are unrelated to relapses. This embodies a phenomenon of “unmasking”: therapies currently in use are excellent at reducing events strictly related to neuroinflammation, which leads to defining the remaining progression events as dependent on other pathways, especially neurodegenerative silent progression.
Comment 2: Mitochondrial dysfunction as a putative mechanism underlying PIRA should be mentioned.
We appreciate and acknowledge the reviewer's observation, allowing us to include this interesting pathological mechanism in our review.
We have added a paragraph, and relative references, in section 3: Mitochondrial dysfunction has been also pointed out as a putative mechanism underlying PIRA. The development of the neurodegeneration molecular cascade is significantly influenced by nuclear factor erythroid 2-related factor 2 (Nrf2)-impaired signaling, fundamental to prevent oxidative stress and mitochondrial dysfunction and found to be compromised in MS [33, 34]. It is possible for mitochondrial disfunction to be triggered by reactive oxygen species (ROS) produced by innate immune cells, spurring the process of axonal swelling and fragmentation in disease progression [6, 35]. Furthermore, both reactive astrocytes and chronically demyelinated axons have a higher quantity of mitochondria because demyelination raises the energy requirement to maintain a proper intra-axonal ion balance [36].
We have also discussed this topic in the biomarker section: Compared to non-inflammatory control patients, subjects with progressive MS had higher CSF concentrations of free circulating mitochondrial DNA (mtDNA) copies, which resulted positively correlated with EDSS worsening and a higher T2 lesion load [84]. Therefore, elevated levels of cell-free mtDNA resulted linked to disease progression mechanisms, stressing the key role of mitochondria. CSF levels of lactate are also related to mitochondrial dysfunction and were linked to an increase of neurological impairment and to levels of both tau and neurofilament light protein levels [85].
Comment 3: Given that aging has a major impact on the pathological and immunological processes in MS and that there has been a recent shift in MS prevalence towards older age and also a marked increase in the proportion of patients with late-onset MS, it would be useful to present data on PIRA in late-onset MS, if such data exist.
We thank the reviewer for this comment which enable us to better clarify this topic. A previous study of our group (Iaffaldano P, Portaccio E, Lucisano G, et al. Multiple Sclerosis Progression and Relapse Activity in Children [published correction appears in JAMA Neurol. 2024 Jan 1;81(1):88]. JAMA Neurol. 2024;81(1):50-58) used data of the Italian Multiple Sclerosis Register to assess the incidence of and factors associated with PIRA and RAW in POMS compared with typical adult-onset MS (AOMS) and late-onset MS (LOMS). Factors associated with PIRA were older age at onset (AOMS vs POMS HR, 1.42; 95% CI, 1.30-1.55; LOMS vs POMS HR, 2.98; 95% CI, 2.60-3.41; P < .001), longer disease duration (HR, 1.04; 95% CI, 1.04-1.05; P < .001), and shorter DMT exposure (HR, 0.69; 95% CI, 0.64-0.74; P < .001). The incidence of PIRA was 1.3% at 20 years of age, but it rapidly increased approximately 7 times between 21 and 30 years of age (9.0%) and nearly doubled for each age decade from 40 to 70 years (21.6% at 40 years, 39.0% at 50 years, 61.0% at 60 years, and 78.7% at 70 years). We included the following specification in the first section of the manuscript: A longer disease duration and an older age at onset were factors linked to PIRA: between the ages of 21 and 30, the frequency of PIRA almost doubled, and between the ages of 40 and 70, it increased by around seven times in a recent study [15]. The effect of PIRA in accelerating EDSS progression is less pronounced in pediatric-onset MS than in adult-onset MS and late-onset MS, as outlined in another study of our group cited in the text (Simone M, Lucisano G, Guerra T, et al. Disability trajectories by progression independent of relapse activity status differ in pediatric, adult and late-onset multiple sclerosis. J Neurol. 2024;271(10):6782-6790). We included in the text: Compared to patients without PIRA, individuals with PIRA experience a noticeably higher rise in EDSS scores, and PIRA's impact on hastening the progression of EDSS is less no-ticeable in pediatric-onset MS than in adult-onset and late-onset MS [17]. Challenges of treatment for late-onset MS patients have been outlined also in a recent paper by Tunc et al (Tunç A, SeferoÄŸlu M, Sivaci AÖ, Köktürk MD, Polat AK. Pediatric, adult, and late-onset multiple sclerosis patients: A unified analysis of clinical profiles and treatment responses. Mult Scler Relat Disord. Published online November 22, 2024), and we included a short citation in the last section of the manuscript: Patients with onset at different age have very deeply heterogeneous clinical profiles and outcomes, which highlights the need for “age-specific” MS management approaches, especially for late-onset patients [123].
Comment 4: It would also be useful to explain the term PIRMA.
We thank the reviewer for this suggestion, useful to expand, in our manuscript, the discussion about the challenging definition of PIRA. PIRMA definition has been added in the Introduction section as follows “Recently, the definition of PIRMA, progression independent of relapse and MRI activity, was also coined to expand the clinical definition of PIRA including the absence of MRI activity between three months after and one month before the onset of a relapse”. In the final section of the manuscript, we also addressed this interesting topic. There are many challenges in the field of neuroimaging, to better characterize the processes underlying the definition of PIRA and to understand the phenomena of neurodegeneration. The term "advanced-PIRMA" has been recently proposed by Ciccarelli et al. to stress the advocacy of employing both conventional and advanced imaging assessment to examine the contributions to disability accumulation of all the underlying pathologic processes, considering new lesions, PRLs, SELs, atrophy and all neurodegenerative radiological parameters [13].
Reviewer 2 Report
Comments and Suggestions for Authors
Tommaso. et al. Progression of MS results in disability accumulation independently of both relapsing activity (PIRA) and relapse-related deterioration, mechanisms involving ongoing interactions between neuroinflammation and neurodegeneration in the brain. Moreover, the result is technically sounded and worthy to be published in Int. J. Mol. Sci.
The following are some comments and suggestions that are given to improve the manuscript:
Comment 1: What are the specific aspects of the interaction between PIRA and recurrent disease course?
Comment 2: Does improving lymphatic cerebrospinal fluid system function delay or reverse non-relapse-related progression of MS?
Comment 3: Are there new imaging techniques that allow more precise monitoring of disease activity and neurodegenerative changes?
Comment 4: How can MS registries be more effectively integrated with genetic and histopathological databases?
Author Response
Response to Reviewer 2 Comments
The reviewer's time spent reviewing this manuscript is greatly appreciated. Please find the detailed responses below and the corresponding revisions highlighted in track changes in the re-submitted files.
Comment 1: What are the specific aspects of the interaction between PIRA and recurrent disease course?
We thank the reviewer for this comment which enable us to better clarify this topic. PIRA is the major contributor to the disability accumulation throughout the entire disease course, and we outlined this interaction in the Introduction section. The impact of age in this complex interplay is crucial: we have included a specification about the correlation between PIRA, age and recurrent disease course, better characterizing two previous work of our group cited in the text. A longer disease duration and an older age at onset were factors linked to PIRA: between the ages of 21 and 30, the frequency of PIRA almost doubled, and between the ages of 40 and 70, it increased by around seven times in a recent study [15]. Compared to patients without PIRA, individuals with PIRA experience a noticeably higher rise in EDSS scores, and PIRA's impact on hastening the progression of EDSS is less noticeable in pediat-ric-onset MS than in adult-onset and late-onset MS [17]. Moreover, the impact of PIRA and clinical measures in the recurrent course is also outlined in a study by Cagol et al. (Cagol A, Schaedelin S, Barakovic M, et al. Association of Brain Atrophy With Disease Progression Independent of Relapse Activity in Patients With Relapsing Multiple Sclerosis. JAMA Neurol. 2022;79(7):682-692), and we included in the text this specification, and its relative reference: Patients with a relapsing course and PIRA also experience rapid brain atrophy, especially in the cerebral cortex [20]. Challenges in understanding, measuring and treating PIRA have been also better highlighted in the final section of the review.
Comment 2: Does improving lymphatic cerebrospinal fluid system function delay or reverse non-relapse-related progression of MS?
We greatly thank the reviewer for this thoughtful question, allowing us to include this topic in the review. In the first place, also in line with the comments of other reviewers, we discussed better the role of the glymphatic system, in the sections 3 and 4.
A “neuro-lympho-vascular component” in neurodegenerative conditions has been proposed in the last year. (Louveau A, Da Mesquita S, Kipnis J. Lymphatics in Neurological Disorders: A Neuro-Lympho-Vascular Component of Multiple Sclerosis and Alzheimer's Disease?. Neuron. 2016;91(5):957-973. doi:10.1016/j.neuron.2016.08.027). Starting with this statement, we included in the treatment section a recent study by das Neves et al. (das Neves SP, Delivanoglou N, Ren Y, et al. Meningeal lymphatic function promotes oligodendrocyte survival and brain myelination. Immunity. 2024;57(10):2328-2343.e8). A “neuro-lympho-vascular component” in neurodegenerative conditions has been proposed in the last year [113]. A recent study of das Neves et al. highlighted that reduced vascular endothelial growth factor C in the CSF can be observed in MS patients, especially soon after clinical relapses, constituting a possible sign of impaired meningeal lymphatic function. In light of this and the fact that meningeal lymphatics control oligodendrocyte function and brain myelination, this study found that inducing ablation of the meningeal lymphatic vessels in adult mice resulted in changes in glial cell gene expression [114].
Comment 3: Are there new imaging techniques that allow more precise monitoring of disease activity and neurodegenerative changes?
We thank the reviewer for this comment, which allowed us to expand the discussion of radiological challenges in PIRA assessment. In first place, in line with other reviewers’ suggestions, we have added in the manuscript text the definition of PIRMA. Recently, the definition of PIRMA, progression independent of relapse and magnetic resonance imaging (MRI) activity, was also coined to expand the clinical definition of PIRA including the absence of MRI activity between three months after and one month before the onset of a relapse [13].
Compartmentalized meningeal inflammation is thought to represent one of the key players in the pathogenesis of cortical demyelination in MS. Therefore, we included in the original manuscript results of Herranz et al. (Brain, 2024), which highlighted the role of TSPO-PET in MS for imaging in vivo inflammation in the cortico-meningeal brain tissue compartment, providing in vivo evidence of meningeal inflammation. We have stressed this concept in the manuscript: Considering the key role of meningeal inflammation in MS pathogenesis and progression mechanisms, findings of Herranz et al. strongly support the role of TSPO-PET for imaging in vivo cortico-meningeal inflammation [51].
We have also highlighted new MRI techniques which depict the role of glymphatic system impairment on neurodegeneration in MS: A reduced diffusion tensor imaging along the perivascular space (DTI-ALPS) index has been recently correlated to SPMS conversion, focusing on the possible key role of micro-structural glymphatic disruptions in MS progression mechanisms [52].
In the final section, we have also included a short paragraph about the challenges of neuroimaging assessment in PIRA evaluation. We have also cited the possibilities of machine learning and artificial intelligence approaches. There are many challenges in the field of neuroimaging, to better characterize the processes underlying the definition of PIRA and to understand the phenomena of neurodegeneration. The term "advanced-PIRMA" has been recently proposed by Ciccarelli et al. to stress the advocacy of employing both conventional and advanced imaging assessment to examine the contributions to disability accumulation of all the underlying pathologic processes, considering new lesions, PRLs, SELs, atrophy and all neurodegenerative radiological parameters [13]. Artificial intelligence and machine learning approaches have the potential to improve MS research: by combining data from many sources, these methods might predict clinical impairment and future disease progression, generating also measures that were previously only achievable with specialized imaging (such as synthesizing new images from conventional sequences) [119].
Comment 4: How can MS registries be more effectively integrated with genetic and histopathological databases?
We thank the reviewer for this question, which offers an interesting and topical discussion in the field of data collection and integration for Registries. In our experience, the possible promoters of this integration are precisely studies and projects of studies, which aim to collect transversal data, not only clinical, on patients. The Italian MS Society supported several data-sharing initiatives in the last years related to the Register, such as the PROgnostic GEnetic factors in Multiple Sclerosis (PROGEMUS). In 2023, the project BARCONDING MS aimed to build an integrated network of clinical, epidemiological, imaging, genetic data, and also patient-reported outcomes, through the connection of different databases.
We have added a specification, and the relative reference, in the last section of the manuscript. Therefore, the call for studies aimed at collecting clinical, genetic, and imaging data together is crucial, encouraging the idea of integrating datasets to define a multidimensional picture of MS patients [122]. (Zaratin P, Samadzadeh S, SeferoÄŸlu M, et al. The global patient-reported outcomes for multiple sclerosis initiative: bridging the gap between clinical research and care - updates at the 2023 plenary event. Front Neurol. 2024;15:1407257. Published 2024 Jun 20. doi:10.3389/fneur.2024.1407257)
Reviewer 3 Report
Comments and Suggestions for Authors
The review article entitled “A Window to New Insights on Progression Independent of Relapse Activity in Multiple Sclerosis: Role of Therapies and Current Perspective” investigates the neuropathological, immunological and pharmacological features of MS. It is a well-structured article with specific information about multiple sclerosis. There are some points to improve the quality of this article before accepting for publication:
1- High level of similarity reaches to more than 33%. Authors need to rephrase their article to reduce the level of similarity
2- Methodology is messing important points like:
-The time frame covered by this review.
-Quality assessment methods for the selected articles
-Followed regulations to avoid bias in the different stage of this review: planning, data collection, analysis…..etc
3- The authors referred to the importance of regulatory molecules and their associated receptors in peripheral T-cell tolerance, T-cell function however, they didn’t explain these molecules. Write a short paragraph explaining the nature and types of these molecules and their receptors.
4- References are written with a wrong format go to author guidelines and follow the correct format
5- Put the reference’s number before the full stop not after through the whole manuscript
Author Response
Response to Reviewer 3 Comments
The reviewer's time spent reviewing this manuscript is greatly appreciated. Please find the detailed responses below and the corresponding revisions highlighted in track changes in the re-submitted files.
Comment 1: High level of similarity reaches more than 33%. Authors need to rephrase their article to reduce the level of similarity
We thank the reviewer for the careful scrutiny of the manuscript. In line with this thoughtful suggestion, we have rewritten several parts of the article, in all sections. Changes are tracked in yellow in the "tracked changes" manuscript version. The revised version of the Manuscript, without tracked changes, is also uploaded.
Comment 2: Methodology is missing important points like:
-The time frame covered by this review.
-Quality assessment methods for the selected articles
-Followed regulations to avoid bias in the different stage of this review: planning, data collection, analysis…..etc
We greatly thank the reviewer for this thoughtful suggestion, which allows us to improve the quality of our manuscript. We completely rewrote the Methodology paragraph, following the reviewer’s suggestions.
Methodology
2.1 Aims and Research Planning
This review aimed to provide a comprehensive and up-to-date overview of the neuropathological evidence related to the mechanisms underlying PIRA phenomena in MS, with particular reference to studies investigating the impact of currently available DMTs on these phenomena. We carefully scrutinized the MS literature, examining and discussing the numerous neuroimmunology topics in the different review thematic sections.
2.2 Search strategy and inclusion and exclusion criteria
A specific approach for selecting literature sources has been applied in the two primary academic databases, PubMed and Google Scholar. Our search strategy comprised pertinent keywords related to PIRA and smouldering disease in MS. During the selection procedure, particular inclusion and exclusion criteria were applied to guarantee the high standard and applicability of this review. We considered: the most recent PIRA-focused research articles or reviews; studies offering insights into pathophysiology and immunology implications of MS and related to disease progression mechanisms; clinical trials, studies of disease registers and real-world cohorts on therapeutic targets in MS in terms of neuroinflammation and neurodegeneration. In two cases, we also included topics discussed at the most important European congress on MS research, the European Committee for Treatment and Research in Multiple Sclerosis. Following this search path, some papers were eliminated since they were either unpublished manuscripts or non-peer-reviewed materials. Specifying the time frame of the search, no restrictions were placed on publication dates, but we focused mainly on studies published in the last five years. Thus, about 85% of citations retrieved are from studies published after 2020, and about 32.5% are research published in 2024.
Comment 3: The authors referred to the importance of regulatory molecules and their associated receptors in peripheral T-cell tolerance, T-cell function however, they didn’t explain these molecules. Write a short paragraph explaining the nature and types of these molecules and their receptors.
We thank the reviewer for this comment, which allowed us to expand this topic. We have rewritten the paragraph as follows, including specifications about T-cell tolerance mechanisms, in the section 6 “Adaptive immunity and PIRA: Role of T cells and B cells”
The intricate interplay between extrinsic mechanisms, requiring regulatory T cells (Tregs), and T-cell specific costimulators regulates peripheral tolerance. The importance of regulatory molecules and their associated receptors in peripheral T-cell tolerance, T-cell function, and their activation pathways in MS have been extensively outlined in a recent review [67]. MS has been linked to disruptions in the expression of various costimulatory signaling molecules necessary for T-cell activation [68]. The interaction of programmed cell death 1 (PD-1) and its ligands results in the start of inhibitory signals that control tis-sue damage and T-cell activation [69]. It has been confirmed that the PD-1/PD-L1 axis is a classical immune suppressor in MS, and despite the regulation mechanisms of this axis throughout the MS course are still unknown, several studies have shown PD-L1 deficiency and a correlation with disease progression [70]. CD137 has been proposed to be involved in defective regulatory function of Treg and dendritic cells and resulted elevated in MS individuals, while FoxP3 was found to be impaired in relapsing phenotypes [68]. B7H4, VISTA, CTLA-4, BTLA, Lag-3 and TIM3 receptors and their ligands have also a key role in peripheral tolerance and an inhibitory effect on cytokine production [67].
Comment 4: References are written with a wrong format go to author guidelines and follow the correct format
Put the reference’s number before the full stop not after through the whole manuscript
We thank the reviewer for this observation. We have revised the references, in accordance with the Author’s guideline, and inserted the number of the citation before the full stop in the whole manuscript.
Reviewer 4 Report
Comments and Suggestions for Authors
This is a detailed yet concise review of the neuropathological and immunological evidence related not only to relapse-related deterioration in multiple sclerosis, but also to the newer concept progression independent of relapse activity (PIRA). In addition to describing the biology of relapses in multiple sclerosis, radiological indicators and molecular features of inflammatory processes, the role of T and B cells, and biomarkers potentially related to PIRA phenomena, a particular contribution of the work is an overview of disease-modifying therapies and PIRA. I believe that this clearly presented and well-written paper makes an important contribution to the current field.
- The main question addressed by the research is an overview of the neuropathological and immunological evidence related to the progression independent of relapse activity (PIRA) phenomena in multiple sclerosis (MS), with a particular focus on the impact of available therapies on these complex mechanisms.
- The topic is current and relevant to the field as it brings together recent findings on smouldering MS activity (caused by acute peripheral inflammation, chronic neuroinflammation and neurodegeneration) and the relatively recent PIRA phenomenon, as well as the diagnostic criteria for MS proposed at ECTRIMS 2024. It also fills a gap in the field of currently available therapies and their impact on PIRA.
- The paper provides a comprehensive overview of the PIRA phenomenon mentioned above, biological findings on relapses in multiple sclerosis, radiological indicators and molecular characteristics of inflammatory processes, the role of T and B cells. It also provides (add) insights into the correlation of CSF biomarkers and disease-modifying therapies with PIRA.
- The conclusions are consistent with the evidence presented alhough the glymphatic function in MS could be explained in some previous section. The authors conclude that novel perspectives on the biology and progression of MS have helped to promote and confirm the idea of a smouldering MS. In this section, the authors focus more on future directions and perspectives, as the subtitle "Conclusions and Future Perspectives" suggests.
- The references are up-to-date, numerous and important to the field, although they contain at least 6 self-citations. I would say that they are appropriate.
- The paper contains one Figure and one Table. It is possible that further figures would aid understanding of the mechanisms, but given what is clearly written, this is not necessary. The table clearly shows the previous studies on the effect of disease-modifying therapies on PIRA in different patient groups and their highlighted results.
Author Response
Response to Reviewer 4 Comments
The reviewer's time spent reviewing this manuscript is greatly appreciated. Please find the detailed responses below and the corresponding revisions highlighted in track changes in the re-submitted files.
Comment 1: The conclusions are consistent with the evidence presented although the glymphatic function in MS could be explained in some previous section.
We sincerely thank the reviewer for the scrutiny of our review and for the appreciation of the intentions of our research work and the discussion of the topics covered. Thank you for pointing this out: in line with the reviewer’s suggestion, we expanded the discussion about the impairment of the glymphatic system in progression mechanisms.
In the section 3 “Smouldering biology in multiple sclerosis”, we included the following sentences: Abnormalities in the glymphatic system, a complex CNS ‘waste clearance’ system, have been reported in several neurodegenerative diseases, including MS. Impaired glymphatic function in MS was associated with measures of neurodegeneration and demyelination, in a proposed mutual relation: the glymphatic system dysregulation exacerbates MS symptoms and, in parallel, MS neuroinflammation impacts the correct functioning of the system [37, 38]. Mechanisms of inflammation and demyelization can disrupt astrocyte function, essential to glymphatic system activity [39]. One of the connections between MS and glymphatic dysfunctions may be also the presence of inflammatory lesions in the perivenous spaces, a drainage route through which the glymphatic system eliminates waste products [38]. Carotenuto et al. [37] demonstrated that the glymphatic function was usually compromised in MS patients when compared to healthy controls, and PPMS patients showed the most severe impairment.
In the fourth section, we included this specification: A reduced diffusion tensor imaging along the perivascular space (DTI-ALPS) index has been recently correlated to SPMS conversion, focusing on the possible key role of micro-structural glymphatic disruptions in MS progression mechanisms [52].
Finally, in the last section, we stressed the importance of this topic: Since the glymphatic system's role in MS progression is still a relatively young field of study, additional research is required to fully comprehend the nature of this mutual relationship and any potential therapeutic applications [37, 38].
Finally, we appreciate and thank the reviewer for the comments about the clarity of the text, figure and table.
Round 2
Reviewer 2 Report
Comments and Suggestions for Authors
The author has answered all the questions.
Reviewer 3 Report
Comments and Suggestions for Authors
All the comments were addressed by the authors, and I approve the manuscript for publication.